# Tissue-specific and convergent metabolic transformation of cancer correlates with metastatic potential and patient survival

Edoardo Gaude[1] & Christian Frezza[1]

Cancer cells undergo a multifaceted rewiring of cellular metabolism to support their biosynthetic needs. Although the major determinants of this metabolic transformation have been elucidated, their broad biological implications and clinical relevance are unclear. Here we systematically analyse the expression of metabolic genes across 20 different cancer types and investigate their impact on clinical outcome. We find that cancers undergo a tissue-specific metabolic rewiring, which converges towards a common metabolic landscape. Of note, downregulation of mitochondrial genes is associated with the worst clinical outcome across all cancer types and correlates with the expression of epithelial-to-mesenchymal transition gene signature, a feature of invasive and metastatic cancers. Consistently, suppression of mitochondrial genes is identified as a key metabolic signature of metastatic melanoma and renal cancer, and metastatic cell lines. This comprehensive analysis reveals unexpected facets of cancer metabolism, with important implications for cancer patients' stratification, prognosis and therapy.

[1] MRC Cancer Unit, University of Cambridge, Hutchison/MRC Research Centre, Box 197, Cambridge Biomedical Campus, Cambridge CB2 0XZ, UK. Correspondence and requests for materials should be addressed to C.F. (email: cf366@MRC-CU.cam.ac.uk).

Cancer has been defined as a genetic disease whereby the evolution from benign to malignant lesions occurs via a series of mutations over time[1]. The process of transformation is accompanied by profound alterations of cellular metabolism that fulfil the energy requirements of cancer cell growth and proliferation[2]. Dysregulation of cellular metabolism in cancer cells was originally described by Otto Warburg almost a century ago[3]. He observed that metabolism of cancer cells relies mostly on glycolysis even in the presence of oxygen, whereas normal cells fully oxidize glucose in the mitochondria. These findings remained partially neglected until recently, when the availability of state-of-the-art technologies enabled a more comprehensive examination of the intricacies of cancer metabolism. It is now apparent that the metabolic reprogramming of cancer goes beyond activation of glycolysis. For instance, a recent systematic analysis of expression of metabolic genes across several cancer types showed that, besides glycolysis, other metabolic pathways, including nucleotides and protein synthesis, are activated in cancer[4]. In support to an increased requirement of building blocks for nucleotide biosynthesis, Jain and colleagues found that increased glycine uptake strongly correlates with proliferation rates of cancer cells from the NCI-60 database[5].

Although these metabolic features of cancer are now exploited for diagnostic and therapeutic purposes, their broader clinical implications are still under intense investigation. In this study we analyse expression data from 20 different solid cancers, encompassing a total of 8,161 cancer and normal samples from TCGA database to comprehensively investigate the metabolic transformation of cancer and its implications for patient prognosis. Consistent with previous observations[4], we show that these cancers exhibit common metabolic signatures, but maintain some features of their tissue of origin. Importantly, by distinguishing tissue-dependent and tissue-independent metabolic signatures, we find that activation of nucleotide synthesis and inhibition of mitochondrial metabolism are main features of the convergent metabolic landscape of cancer. Furthermore, we find that downregulation of oxidative phosphorylation correlates with poor clinical outcome across several cancer types and it is associated with the presence of epithelial-to-mesenchymal (EMT) signature. Consistently, loss of oxidative phosphorylation (OXPHOS)-related genes is observed in metastatic melanoma samples, compared to the respective primary tissue. Overall, our analysis reveals novel and clinically relevant aspects of the metabolic transformation of cancer, with important implications for patient stratification, prognosis and therapy.

## Results

**The metabolic landscape of cancer.** In order to investigate the metabolic landscape of cancer, we analysed the expression of metabolic gene across 20 different types of solid cancers from TCGA, encompassing a total of 8161 cancer and normal samples (Supplementary Table 1 and Supplementary Fig. 1 for a schematic of the pipeline). RNAseq data from each cancer data set were analysed using a negative binomial generalized linear model (see Methods and ref. 6), comparing the expression of metabolic genes in cancer tissues against tissues of origin (Supplementary Table 2). Gene Set Enrichment Analysis (GSEA)[7] was then applied against a manually curated metabolic gene signature (Supplementary Table 3 and Methods for details on the process). While composing metabolic gene signatures we noticed that several genes (~20%) were associated with multiple metabolic pathways (Supplementary Table 4), in line with an interconnected topography of the metabolic network. We reasoned that

promiscuity of genes across metabolic pathways can be a confounding factor when linking differential expression of a gene to a specific function. Indeed, in some cases significant enrichment of metabolic pathways was driven by promiscuous genes only, even without changes in pathway-specific genes (Supplementary Fig. 2). To account for this factor, we applied a correction for gene promiscuity in metabolic pathways (Supplementary Fig. 2 and Methods). Promiscuity-corrected differential gene expression between cancer and normal tissues was then subjected to GSEA and significantly enriched metabolic pathways for each cancer type were obtained (Fig. 1a, Supplementary Fig. 2 and Supplementary Table 5).

We then searched for metabolic pathways that are differentially regulated in more than 25% of cancers compared to corresponding normal tissues (Fig. 1b). Besides glycolysis, a well-established metabolic feature of cancer, purine biosynthesis and DNA synthesis were the most frequently upregulated pathways across different cancers (14/20, 70% and 10/20, 50%, respectively). Phosphoribosylaminoimidazole carboxylase, phosphoribosylaminoimidazole succinocarboxamide synthetase (PAICS) was the most frequently upregulated gene (71%) within the purine biosynthesis pathway. Of note, purine biosynthesis and PAICS expression exhibited strong positive correlation with growth rate of the NCI-60 panel of cancer cell lines (Supplementary Fig. 3), confirming the relevance of this pathway for cancer cell proliferation. Another shared metabolic feature of cancers that emerged from this analysis is the dysregulation of genes encoding for mitochondrial metabolism (Fig. 1a-b). Overall, 65% of cancers exhibited downregulation of at least one mitochondrial pathway, while the remaining 35% showed its over-expression (Supplementary Table 5). In particular, downregulation of Citric Acid Cycle (CAC) and mitochondrial fatty acids oxidation (FAO) genes was observed in 40% and 30% of cancer types, respectively. OXPHOS was found upregulated in 35% and downregulated in 25% of cancers (Fig. 1a-b), showing heterogeneous distribution across cancers.

To validate these findings we took advantage of a recently published study where gene expression and metabolite abundance were measured in a cohort of breast cancer patients[8]. First, we wanted to assess whether expression of metabolic genes correlates to expected changes in metabolite concentration. Expression levels of glycolytic genes positively correlated with accumulation of lactate (Supplementary Fig. 4a), and expression of purine biosynthesis and DNA synthesis correlated with abundance of nucleotides (Supplementary Fig. 4b-c). Moreover, gene expression of FAO negatively correlated with palmitate levels (Supplementary Fig. 4d). We then applied metabolic GSEA on these cancer samples. Among metabolic pathways enriched between breast cancer and normal samples, purine biosynthesis and DNA synthesis were upregulated, while CAC, FAO and cyclic nucleotides metabolism were downregulated (Fig. 1c), thus confirming our findings with an independent and cross-platform data set.

**Tissue-specific features of cancer metabolism.** When performing hierarchical clustering of enriched metabolic pathways, we observed that cancers arising from the same tissue, or anatomically related sites, exhibit similar metabolic features (Fig. 1a). Notably, cancers maintained tissue-specific metabolic signatures even when analysed independently from their tissue of origin (Supplementary Fig. 5). To corroborate this observation, we performed correlation analysis between the metabolic signatures of distinct cancers and corresponding normal tissue (Fig. 2a and Supplementary Table 6). Most correlations were positive and significant (57/96, Spearman ρ, Benjamini-Hochberg

adjusted P-value <0.05), confirming that the metabolic landscape of cancer is reminiscent of its tissue of origin. Interestingly, we also observed few significant negative correlations (4/96), including highly expressed pathways in normal tissues that were downregulated in cancer. The overall loss of tissue-specific metabolic functions and the convergence to a common metabolic landscape across cancers was confirmed by the finding that the variance of metabolic pathways among cancers was lower than the variance among normal tissues (Fig. 2b).

To further investigate tissue-specific metabolic rewiring of cancer, we first identified metabolic pathways that are enriched in each normal tissue, compared to average (Supplementary Table 7). We then determined the extent of tissue-specific metabolic rewiring in cancers by assessing whether metabolic pathways that characterize a normal tissue change in cancer tissue. While most tissue-specific metabolic functions were not altered in cancer (Fig. 2c), 38% of the metabolic pathways that were highly expressed in normal tissue were downregulated in cancer. Also, 22% of the downregulated pathways in normal tissues were upregulated in cancer (Fig. 2c). Besides the definition of tissue-dependent metabolic pathways, this analysis allowed us also to define tissue-independent metabolic rewiring of cancer.

Notably, purine and pyrimidine biosynthesis and DNA synthesis were among the most commonly (>20%) upregulated pathways, whereas CAC, mitochondrial FAO and urea cycle were the most frequently downregulated ones (Fig. 2d).

We then wanted to investigate whether the observed metabolic rewiring of cancer generates tissue-specific metabolic liabilities. To this aim we took advantage of a recently published RNA interference screening on a large panel of genomically character-ized cancer cell lines (Achilles 2.4)[9]. In line with a tissue-specific metabolic reprogramming of cancer, tissue of origin predicted differential essentiality of 59% of metabolic genes (349/595, analysis of variance-adjusted P-value <0.05). Interestingly, purine biosynthesis and DNA synthesis were among the top predicted functions for tissue-independent essentiality (Supplementary Table 8).

**OXPHOS is linked to clinical outcome and metastasis.** We then investigated whether the observed metabolic alterations correlate with the clinical outcome of cancer patients. To this aim, we took advantage of survival data collected by TCGA. Patients from each cancer type were divided into 'high survival' and 'low survival' groups (See Methods Section for details and Supplementary

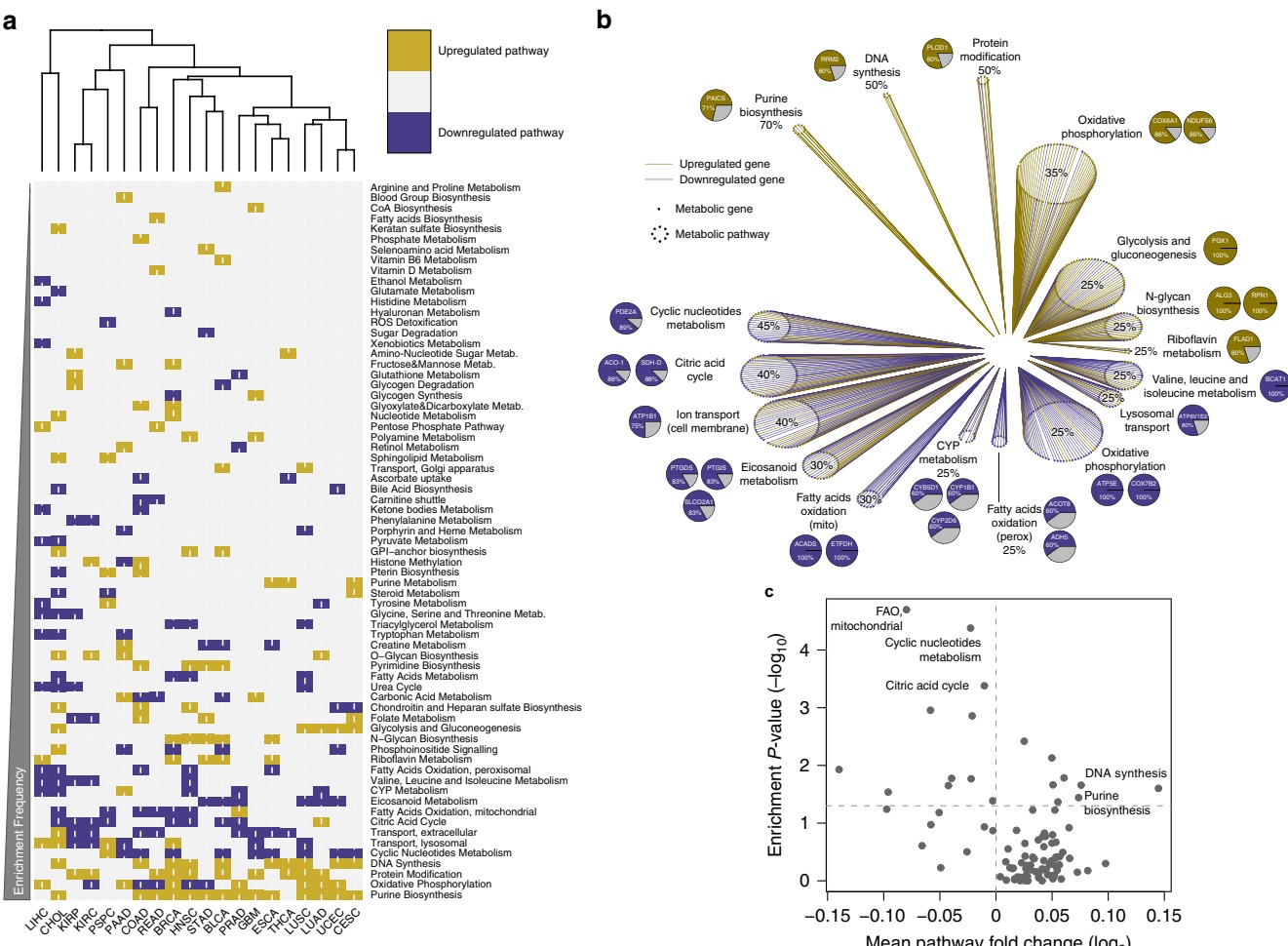

**Figure 1 | Rewiring of metabolic gene expression in cancer tissues compared to normal.** (**a**) Heatmap representation and hierarchical clustering of enriched upregulated (gold) and enriched downregulated (blue) metabolic pathways in cancers compared to normal tissues. (**b**) Gene expression *effect plot* of metabolic pathways enriched in more than 25% of cancers. Circles indicate metabolic pathways and dots in each circle represent individual metabolic genes. Gold and blue lines indicate upregulated and downregulated genes in cancers compared to normal tissues, respectively. Pie charts represent the most frequently up- or downregulated genes in the corresponding pathway; percentage values indicate frequency of up- or downregulation. (**c**) Volcano plot representation of mean fold change expression of genes in each pathway (x axis) vs enrichment P-values (y axis) in breast cancer vs normal samples (data obtained from Terunuma *et al.*[8]). Significantly enriched metabolic pathways in common with Fig. 1b are indicated.

Fig. 6). Then, we performed differential gene expression analysis in low and high survival patients and applied promiscuity-corrected metabolic GSEA. Several metabolic pathways were found significantly altered in the low Survival compared to the High Survival group (Supplementary Table 9). Overall, poor survival was associated with inhibition of at least one mitochondrial pathway in 10/15 cancers (67%). OXPHOS was the most affected pathway in low vs high survival patients and was found downregulated in the low survival group of 9 out of 15 (60%) cancer types (Fig. 3a). The most frequently downregulated genes in this group were subunits of Complex I and IV of the respiratory chain (Supplementary Table 9).

To investigate the possible relation between mitochondrial metabolism and poor clinical outcome, we performed GSEA on low and high Survival patients, taking advantage of a large collection of cancer-associated gene signatures from the Broad Institute. Among cancers that exhibited downregulation of OXPHOS, the most upregulated cellular function was EMT (Fig. 3b), a gene signature associated with cancer aggressiveness

and poor prognosis[10]. Notably, OXPHOS showed significant negative correlation with EMT in 19/20 cancer types (Fig. 3c and Supplementary Fig. 7).

Given the role of EMT in cancer metastasis[10], we hypothesized an association between downregulation of mitochondrial genes, induction of EMT and the metastatic potential of cancer, which is directly linked to patient prognosis. To validate this hypothesis, we took advantage of the Skin Cutaneous Melanoma data set (TCGA), composed of 367 metastatic and 103 primary cancer samples, and performed differential metabolic gene expression and pathway enrichment analyses on metastatic vs primary cancer samples. EMT was strongly upregulated in metastatic vs primary cancer samples (Supplementary Fig. 8a). Furthermore, OXPHOS was the most significantly downregulated metabolic pathway in metastatic vs primary cancers (Fig. 4a). Of note, we could not find significant changes in the expression of the nuclear coactivator PPARγ coactivator-1α (PGC1α), a master regulator of mitochondrial biogenesis previously implicated in cancer metastasis[11,12], between metastatic melanoma vs primary

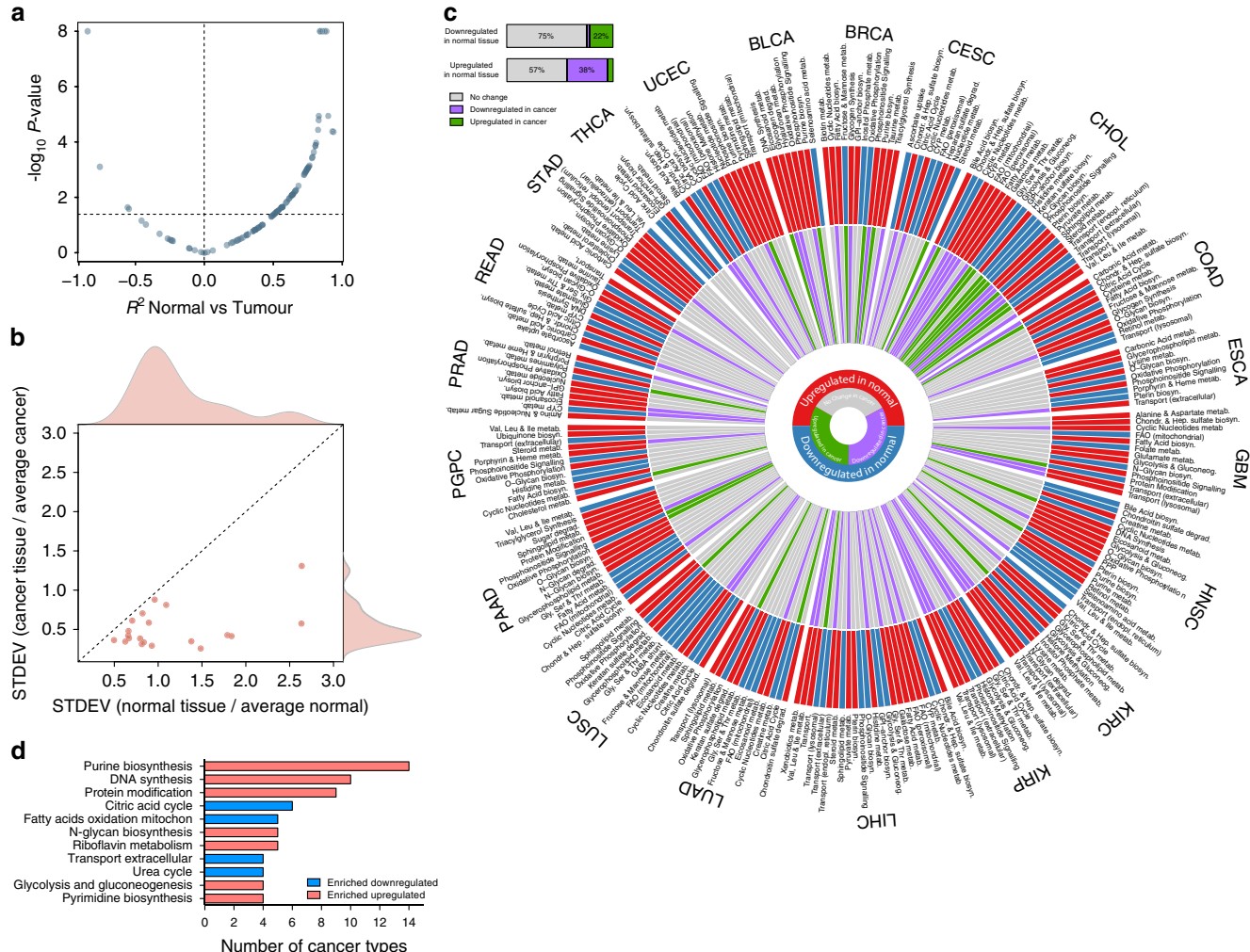

**Figure 2 | The metabolic landscape of cancer is tissue-specific but convergent.** (**a**) Scatter plot representation of correlation coefficient (Spearman, $x$ axis) and correlation $P$-value ($-\log_{10}$, $y$ axis) of metabolic pathways in normal tissue compared to cancer. Horizontal dashed line indicates FDR of 5% ($-\log_{10}$). (**b**) Scatter plot representation of the variance of metabolic pathways among normal ($x$ axis) and cancer ($y$ axis) tissues. (**c**) Tissue-specific metabolic signatures in normal and cancer tissues are represented in a polar histogram. The external circle displays metabolic pathways found enriched upregulated (red) or downregulated (blue) in normal tissues, compared to average. The internal circle shows the enrichment of individual metabolic pathways in cancer compared to normal. Grey bars indicate no change in cancer compared to normal. The horizontal histogram indicates the proportion of metabolic pathways altered in cancer compared to pathways downregulated or upregulated in normal tissues. (**d**) Metabolic pathways enriched in cancer tissue compared to normal, independent of tissue of origin. Metabolic pathways enriched in >20% of cancer types are shown.

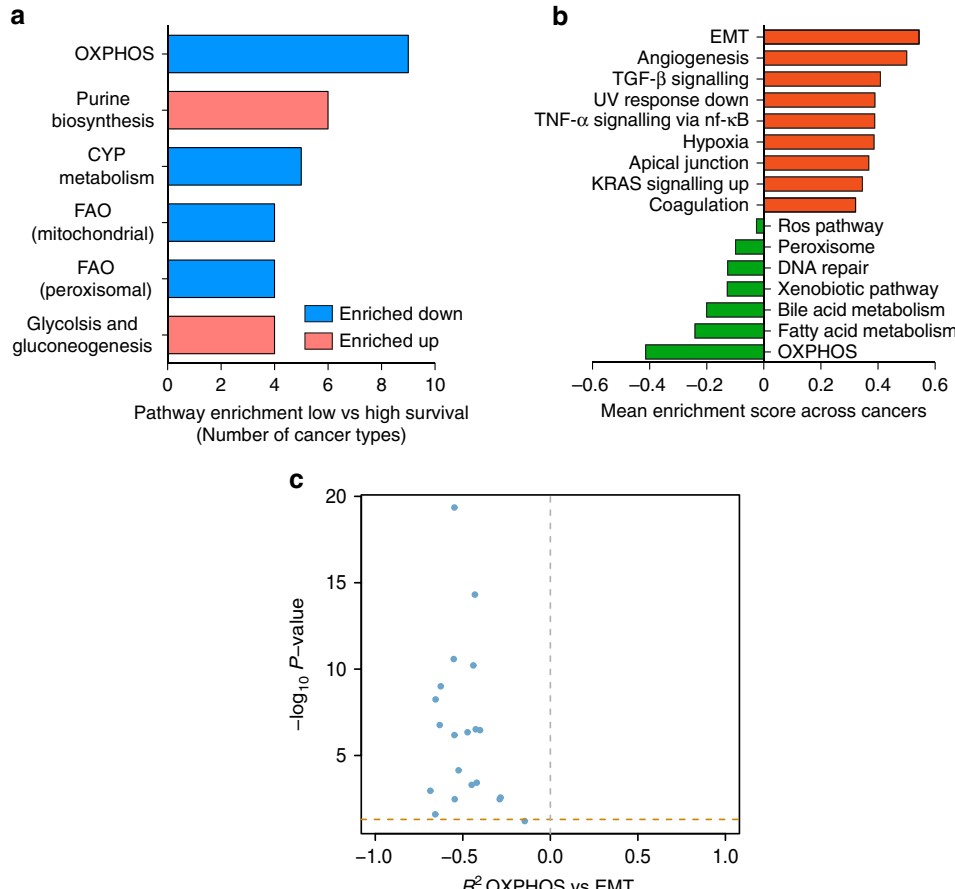

**Figure 3 | Downregulation of OXPHOS genes is associated with poor clinical outcome and EMT gene signature.** (**a**) Frequency of metabolic pathways found enriched upregulated (red) or downregulated (blue) between low and high survival patients in at least 25% of cancer types. (**b**) Top 10 enriched upregulated and downregulated cancer hallmarks between low and high survival patients across cancer types that showed OXPHOS downregulation (9 cancers). Mean enrichment scores of low vs high survival patients across the nine cancer types considered are shown. (**c**) Volcano plot showing correlation coefficient (Spearman, x axis) and correlation P-values (Spearman, − log$_{10}$, y axis) of mean expression of OXPHOS genes compared to mean expression of genes involved in EMT. Horizontal dashed line indicates FDR = 5%.

tumours (BH-P value = 0.37). In line with our findings on low vs high survival patients, cyclic nucleotides metabolism and purine biosynthesis were both upregulated in metastatic vs primary cancers (Fig. 4a). To further validate the link between reduced expression of mitochondrial genes and metastasis, we compared the metabolic gene expression profile of metastatic and parental 786-O kidney cancer cell lines generated by Vanharanta *et al.*[13] In line with our findings in cancer patients, EMT was strongly upregulated in metastatic vs parental cells (Supplementary Fig. 8b) and OXPHOS was the most downregulated metabolic pathway in metastatic cells compared to parental (Fig. 4b). Moreover, cyclic nucleotides metabolism, one of the pathways found upregulated in metastatic vs primary melanoma, was also found upregulated in 786-O metastatic vs parental cell lines (Fig. 4b). Of note, PGC1α levels were not significantly different between the 786-O metastatic vs parental cell lines dataset (BH-P value = 0.51).

Finally, we wanted to assess whether downregulation of mitochondrial gene expression in patients with metastasis is accompanied by changes in metabolite levels. To this aim we took advantage of a recently published study where metabolomics and RNA sequencing were performed on a cohort of 138 clear cell Renal Cell Carcinoma patients[14]. Importantly, downregulation of mitochondrial transcripts was observed in metastatic compared to non-metastatic patients, and it was linked to poor patient survival[14]. Taking advantage of metabolomics data of these

patients we observed that haem and citrate, two metabolites that can only be generated within mitochondria, were among the most downregulated metabolites in metastatic vs non-metastatic patients (Fig. 4c).

## Discussion

Dysregulation of cellular metabolism is now an established feature of cancer. Yet, the contribution of this metabolic reprogramming to cancer biology and to the clinical outcome of patients is still under investigation. Taking advantage of a large collection of cancer samples from TCGA consortium, we systematically investigated the mRNA expression of metabolic genes in 20 different cancer types and assessed the link between altered gene expression and survival of cancer patients. Our analyses revealed that different cancer types exhibited similar metabolic features, which are remnants of their tissue of origin, and that specific metabolic features correlate with metastatic potential and patient prognosis.

Previous studies have highlighted important features of altered metabolism between tumour and normal tissues in a pan-cancer perspective[4,15,16]. For instance, Hu and colleagues performed an extensive analysis of metabolic gene expression changes in cancer compared to normal tissues, observing common patterns of metabolic adaptation among different cancer types[4]. In accordance with this study, we found that distinct cancers display upregulated expression of glycolysis and nucleotide

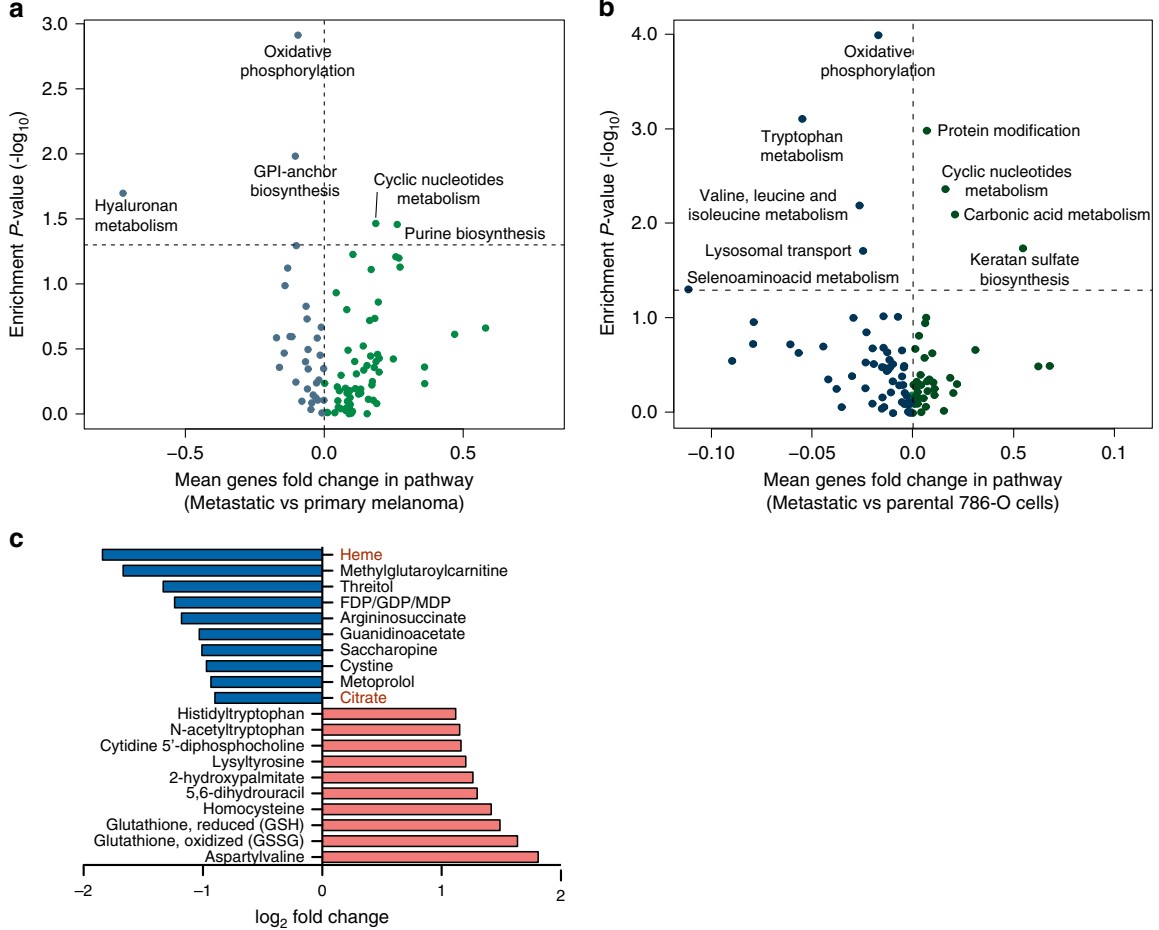

**Figure 4 | Suppression of OXPHOS is a key metabolic feature of Skin Cutaneous Melanoma.** (**a-b**) Volcano plot representation of mean fold change of genes in each pathway (x axis) vs enrichment P-values (y axis) of metastatic vs primary cancer samples (**a**) and of metastatic vs parental 786-O cell lines (**b**). (**c**) Fold changes in metabolite levels of metastatic vs non-metastatic ccRCC patients. Metabolites were ranked according to $\log_2$ fold change. Top 10 downregulated and top 10 upregulated metabolites are shown. Data were obtained from Hakimi et al.[14]

metabolism, and downregulation of FAO, whereas OXPHOS presented heterogeneous regulation. Subtle differences between Hu et al.[4] and our findings, including expression changes of TCA cycle genes, can be explained by differences in the curation of metabolic pathways, which we obtained by integrating multiple databases. Interestingly, the observation that expression of mitochondrial FAO is diminished across several cancer types is in line with the results from a recent pan-cancer analysis where signals from mRNA, miRNA and DNA methylation levels were integrated to find common expression changes in cancer[16].

Together with these findings, our results showed diminished expression of TCA cycle enzymes in cancer, and Succinate Dehydrogenase D (SDHD) ranked among the most frequently downregulated mitochondrial genes, in line with its role as mitochondrial tumour suppressor[17]. This finding is in line with a previous study reporting loss of co-expression of genes of the mitochondrial respiratory chain[15], and with the decrease of mitochondrial DNA (mtDNA) in tumour samples from TCGA database[18]. Together with the observation that direct inhibition of mitochondrial metabolism is responsible for p53 genetic inactivation and increased tumourigenic potential[19], our results support the notion that downregulation of several mitochondrial pathways is a common feature of the metabolic rewiring occurring in different cancer types.

Although these data seem to support a role for mitochondrial dysfunction in cancer initiation and progression, mitochondria are far from being an accessory organelle in cancer cells. Cells completely devoid of mtDNA ($\rho^0$) have lower ability to form tumours in mice[20] and $\rho^0$ cells need to acquire mtDNA from host cells to recover mitochondrial function and achieve growth *in vivo*[21]. Indeed, mitochondria are important for the generation of several precursor molecules, such as aspartate, citrate and succinyl-CoA for supporting nucleotide, lipid and heme biosynthesis, respectively. Moreover, mitochondrial metabolism is flexible and can engage in both oxidative and reductive metabolism to support the generation of cytosolic citrate even in the presence of mitochondrial dysfunction triggered by genetic or environmental cues. For instance, reductive citrate has been shown to support lipid synthesis under hypoxia[22], in the presence of TCA cycle truncation[23], or in the presence of respiratory chain inhibitors[24]. Therefore, without further experimental validation, our results cannot exclude the possibility that partial, rather than complete, loss of mitochondrial function supports the growth of cancer cells by inducing a glycolytic switch, known to support anabolic programmes in fast growing cells[25], while maintaining mitochondrial functions required for metabolism and signalling. Importantly, recent work demonstrated that partial mitochondrial dysfunction induces migration, invasion and metastasis, while complete loss of mitochondrial function leads to inhibition of the metastatic phenotype[26].

Our work established for the first time a link between metabolic alteration and survival of cancer patients.

By comparing low and high survival patients from 15 different cancer types we observed that downregulation of OXPHOS gene expression is almost invariably associated with poor clinical outcome. This result suggests that, despite activation of OXPHOS could have different effects during cancer initiation depending on the tissue of origin, suppression of OXPHOS genes is a common feature of cancer progression and could have important implications for patient survival. Low OXPHOS was strongly associated with induction of EMT, a process linked to cancer invasion and metastasis, and one of the most common causes of cancer deaths. Consistently, OXPHOS was among the most downregulated pathways in distant melanoma metastases, compared to the primary cancer. These results support at much broader scale the finding that partial mitochondrial dysfunction increases metastatic potential of cancer cells[26]. At the same time, these results partially disagree with recent work from the Kalluri's laboratory, where LeBleu and colleagues investigated the metabolic phenotype (MP) of circulating tumour cells and metastasis from various breast cancer models[11]. In accordance with our findings, they found that metastatic cells exhibited low expression of OXPHOS genes, compared to the primary tissue and circulating cancer cells. However, they found that invasive ductal breast cancers are characterized by high expression of the master regulator of mitochondrial biogenesis PGC1α, which also correlated with metastasis. Our analysis did not show significant changes in PGC1α expression in metastatic vs primary cancers, suggesting that the findings of LeBleu are not a common feature of the metabolic transformation of cancer but, likely, apply to a specific subset of breast cancers. In line with the possible tissue-specific role of PGC1α, a recent study found that its downregulation is linked with prostate cancer progression and metastasis, and its genetic reactivation suppresses the formation of prostate cancer metastases[12].

Our analytical approach is not devoid of limitations. First, establishing a link between mRNA levels of metabolic enzymes and cellular function can be a daunting task, not only because of the lack of correlation between transcript abundance and protein concentration[27], but also because of lack of large-scale information about downstream regulation of protein activity (for example, acetylation, phosphorylation, and so on). Moreover, regulation of metabolic pathways can be very intricate and often occurs at nodal points in the pathway, rather than at the level of every gene; therefore, mean expression of metabolic pathways is only a partial estimate of their activity. Second, the association between downregulation of OXPHOS and metastatic behaviour via induction of an EMT signature is based on correlation. Despite this hypothesis is in line with previous studies[26], and we further confirmed such link in an independent data set of metastatic melanoma, more experimental work is required to corroborate the molecular underpinnings linking mitochondrial function to metastasis.

Our results have multiple implications. First, they suggest that to fulfil their metabolic reprogramming cancers explore different molecular paths that entirely depend on the tissue of origin (see Supplementary Fig. 9 for a model). Second, they indicate that, despite the overwhelming genetic complexity that underlines transformation, cancer cells contrive common strategies to support their proliferation. Therefore, we hypothesize that the metabolic reprogramming of cancer is degenerated, that is, different oncogenes and tumour suppressor genes lead to similar metabolic signatures to support proliferation. It is therefore tempting to speculate that evolution of cancer might be driven by phenotypic traits, and that oncogenes and tumour suppressors might be selected for their efficiency in regulating these metabolic changes. In line with this hypothesis, a recent study found that metabolic and cancer-causing genes undergo co-altered somatic copy number variation[28], indicating that alteration of cancer-associated genes is often linked with metabolic rewiring. These findings may catalyse a better understanding of the role of dysregulated metabolism in cancer and provide novel means to stratify patients based on their metabolic features.

## Methods

**Cancer and normal samples selection.** Samples from 20 different solid cancer types were downloaded from The Cancer Genome Atlas data portal (https://tcga-data.nci.nih.gov/tcga/dataAccessMatrix.htm). For each cancer type, level 3 RNAseqV2 Read Counts genes results data of cancer and normal samples were analysed. We considered only normal samples originated from solid normal tissues adjacent, but distal, from the site of tumour. Exact sample sizes of cancer and normal samples used are reported in Supplementary Table 1. *P*-values distributions of each comparison of cancer vs normal obtained from differential gene expression analysis (see below) were considered to check for possible size effects.

**Differential gene expression and pathway enrichment analysis.** Raw counts of RNAseq analysis were obtained from TCGA data base for each cancer data set considered and analysed with the R package DESeq2 (version 1.6.3)[29], which assesses differential gene expression by use of negative binomial generalized linear model, as described by Love *et al.*[29] The outcome of the DESeq analysis (that is, Wald test Statistics of cancer tissue vs normal tissue) was used as an estimate of differential gene expression in the subsequent pathway enrichment analysis. Every gene was associated to one or more metabolic pathways, according to the genome scale metabolic model Recon1 (ref. 30). This metabolic gene signature was then manually curated to include missing genes or functions.

Differential gene expression was corrected for promiscuity across metabolic pathways by dividing the Wald *t*-value statistics obtained from DESeq analysis by the number of associated pathways (promiscuity). Corrected *t*-values were then used as input for GSEA. GSEA was performed by applying the manually curated metabolic gene signature to promiscuity-corrected *t*-values according to the algorithm developed by Subramanian *et al.*[7] by using the R package 'piano' (version 1.6.2)[31].

Validation of the core metabolic signature in primary cancers (Fig. 1) was performed by using gene expression data from Terunuma *et al.*[8], comprising 67 human breast cancer samples and 65 normal tissue controls. Differential gene expression analysis of breast cancer vs normal samples was performed by applying Shapiro Wilk's test for normality followed by two-sided Student's *t*-test and promiscuity-corrected *t*-values were used to perform metabolic GSEA as described above. The same approach was adopted for validation of metabolic adaptation in metastatic 786-O cell lines, compared to parental (data from Vanharanta *et al.*[13], GEO accession code: GSE32299). Metastatic and parental groups were composed of 4 and 3 samples, respectively. These and all subsequent analyses were performed in R software, version 3.1.3 (2015.03.09) 'Smooth Sidewalk'.

**Correlation analyses.** All correlations were calculated using Spearman's method. Final correlation *P*-values were adjusted for multiple testing using Benjamini-Hochberg correction method.

Gene expression data and growth rate values of NCI-60 cancer cell lines were downloaded via CellMiner (http://discover.nci.nih.gov/cellminer/). Correlation between expression of purine biosynthesis and growth rate of NCI-60 cancer cell lines was calculated by comparing mean expression of genes involved in purine biosynthesis pathway and growth rate in each cancer cell line. Correlation between *PAICS* and growth rate was calculated by comparing expression of *PAICS* and growth rate values in each cancer cell line.

Gene expression data and metabolite abundance of breast cancer and normal samples were obtained from Terunuma *et al.*[8] Correlation between expression of metabolic pathways and metabolite abundance was calculated by comparing mean expression of genes and abundance of metabolites involved in each pathway.

Correlation between OXPHOS and EMT levels was determined, for each cancer type, between median expression levels of OXPHOS and EMT genes for high and low survival patients (see above), respectively. EMT gene signature was obtained from the 'Hallmark_Epithelial_Mesenchymal_Transition' gene set (M5930), publicly available at http://www.broadinstitute.org/gsea/msigdb.

**Survival analysis.** Cancer patients from the 20 cancer cohorts that we analysed were divided into 'high survival group' if they have been part of the study and censored 'alive' for an amount of time higher than 75th percentile of total follow-up duration. We included in the 'low survival group' patients that have died during the study within an amount of time lower than 75th percentile of total follow-up duration. For example, in the bladder urothelial carcinoma data set the total duration of the follow-up study is 10.93 years and the 75th percentile observation time corresponds to 1.62 years. We included in the 'High survival' group only patients that have been censored alive for at least 1.62 years, while the 'Low survival' group was composed of patients that have died within the first 1.62 years of the follow-up study. This resulted in a 'High survival' group formed of

61 patients and 'Low survival' group formed of 61 patients as well. Details of each group size are reported in Supplementary Table 9. We excluded from gene expression analysis of low vs high survival patients those cancer types that displayed $n < 5$ in one of the two groups (CHOL, PCPG, PRAD, READ, THCA).

Differential gene expression analysis coupled with GSEA of low survival vs high survival patients was performed as described above. GSEA of hallmarks cellular functions was performed on cancer types that showed downregulation of OXPHOS in low vs high Survival patients. Gene sets were obtained from the 'HALLMARKS' collection of the MSigDB database, publicly available at http://www.broadinstitute.org/gsea/msigdb.

**Tissue-independent metabolic clustering of cancer samples.** In order to perform cancer clustering based on expression of metabolic pathways independently of the tissue of origin, all cancer samples were assembled into a data matrix. RNAseq Raw Counts of metabolic genes of each sample were variance stabilizing transformation normalized, distributed into metabolic pathways according to the metabolic signature described above and mean expression of genes in each metabolic pathway was calculated. Mean expression levels of metabolic pathways for all cancer samples were then subjected to PAM clustering, after estimation of optimal number of clusters via Gap statistic, as described above. Optimal number of clusters estimated was 16. Enrichment of tissues of origin into the 16 MPs was calculated via hypergeometric test (FDR = 0.05) and proportion of samples of each cancer type mapping into each MP was calculated and plotted in Supplementary Fig. 5.

**Analysis of tissue-specific metabolic rewiring.** Samples from all normal tissues and all cancer tissues were grouped and variance stabilizing transformation[32] was applied independently on RNAseq raw counts of metabolic genes belonging to the normal tissues data set and on the cancers data set. For each metabolic gene we calculated the mean expression across patients, in each normal tissue or cancer:

$$\bar{p} = \frac{p_1 + \cdots + p_n}{n} \quad (1)$$

where $n$ is the number of samples in each normal or cancer data set and $\bar{p}$ defines the mean of all patients, for each metabolic gene.

The ratio between expression of each metabolic gene in a tissue and the average expression across all tissues (normal or cancer) was calculated:

$$r_{i,t} = \frac{\bar{p}_{i,t}}{\bar{q}_i} \quad (2)$$

where $\bar{p}_{i,t}$ is the result of equation (1), that is, the average expression of the $i$th metabolic gene in the $t$th tissue (normal or cancer); and $\bar{q}_i$ is the average $\bar{p}_i$ expression across all tissues (normal or cancer). Hence, $r_{i,t}$ defines the fold change, for each gene, between normal (or cancer) tissue and the average of all normal (or cancer) tissues. To find out tissue-specific activation or suppression of metabolic pathways, pathway mean was calculated as follows:

$$S_p = \frac{r_{1i,t} + \cdots + r_{ji,t}}{j} \quad (3)$$

where $j$ is the number of genes in each pathway $p$ and $S_p$ denotes the mean $r$ fold change in the pathway $p$, thus obtaining a fold change of each metabolic pathway in each tissue, compared to average tissue. Given $N_p$ and $C_p$ as the $S_p$ values for normal tissues and cancer tissues, respectively, the correlation between metabolic competence in normal and cancer tissues can be calculated from:

$$G_p = cor(N_p, C_p) \quad (4)$$

where $p$ denotes each pathway.

Metabolic diversity between normal and cancer tissues, compared to average, was quantified by calculating the standard deviation of the $N_p$ and $C_p$ distributions, for each normal and cancer tissue, respectively.

Normal tissue-specific functions were obtained by performing differential gene expression and promiscuity-corrected GSEA of each normal tissue, compared to average. Tissue-specific cancer metabolic adaptation was determined by assessing the enrichment of normal-tissue-specific functions between cancer and corresponding normal tissue. Tissue-dependent and -independent metabolic adaptation of cancer were obtained by extracting metabolic pathways that, if up- or downregulated in normal are up- and downregulated in cancer, and vice versa.

Final gene-level information of Achilles 2.4 shRNA screening was obtained from Cowley et al.[9] and metabolic genes were extracted. Association between gene essentiality and tissue of origin was obtained by using analysis of variance and $P$-values were adjusted using Benjamini-Hochberg method. Adjusted $P$-values lower than 0.05 were used to determine significant associations. To determine tissue-independent pathway essentiality we obtained, for each cell line, a list of essential metabolic genes by extracting the top 5% essential genes, based on ATARIS gene-level score[9]. We then combined cell lines into tissues of origin, thus obtaining a list of essential genes for each tissue. To assess pathway essentiality across different tissues, we measured the occurrence of each essential gene across tissues and calculated the average occurrence per pathway, thus obtaining the mean number of tissues were metabolic genes, in each pathway, are essential.

**Data availability.** The TCGA data referenced during the study are available in a public repository from the TCGA website (https://gdc-portal.nci.nih.gov). The data from Terunuma et al.[8] referenced during the study are available in a public repository from the GEO website (http://www.ncbi.nlm.nih.gov/geo) under the accession number GSE39004/GSE37751. Achilles 2.4 data from Cowley et al.[9] referenced during the study are available in a public repository Figshare website (https://dx.doi.org/10.6084/m9.figshare.1019859). The data from Vanharanta et al.[13] referenced during the study are available in a public repository from the GEO website (http://www.ncbi.nlm.nih.gov/geo) under the accession number GSE32299. The authors declare that all the other data supporting the findings of this study are available within the article and its supplementary information files and from the corresponding author on reasonable request.

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

## Acknowledgements

C.F. and E.G. thank Medical Research Council (MRC) Core funding for financial support. E.G. was supported by MRC Doctoral Training Partnership (DTP) studentship. We thank Dr Evelina Gabasova (MRC Cancer Unit, University of Cambridge, Cambridge, UK) for advice on statistical analyses.

## Author contributions

C.F. and E.G. designed experimental pipeline and discussed results. E.G. performed all bioinformatics analysis. C.F. and E.G. wrote the manuscript.

## Additional information

**Competing financial interests:** The authors declare no competing financial interests.

**How to cite this article**: Gaude, E. & Frezza, C. Tissue-specific and convergent metabolic transformation of cancer correlates with metastatic potential and patient survival. *Nat. Commun.* 7:13041 doi: 10.1038/ncomms13041 (2016).

