## [Peer Review File · Nature Communications]

Reviewers' comments:

Reviewer #1_Cancer Metabolism
(Remarks to the Author):

This paper demonstrates that cancers undergo a tissue-specific metabolic rewiring, which converges on downregulation of mitochondrial genes. This is associated with the worst clinical outcome across all cancer types and correlates with induction of epithelial-to-mesenchymal transition (EMT). I think the paper is done well and very important for the field.

I have one issue with the paper in interpretation.

I don't think it is appropriate to equate down regulation of mitochondrial genes with mitochondrial dysfunction. It is well known that mitochondrial genes/enzymes are in excess. Thus even 50-75% decrease will not impact oxidative phosphorylation. So please remove that language regarding mitochondrial dysfunction.

Also it is important they clearly state that mitochondria can undergo oxidative and reductive metabolism. Another words, mitochondria always have to engaged in cancer for tumorigenesis. After all mitochondria provides citrate, aspartate and succinyl-CoA necessary for lipid, nucleotide and heme synthesis, respectively.

See recent paper in Cell by Ralph Deberardinis.

Given this what is the advantage to down regulate these genes? Maybe to increase or decrease ROS for metastasis?

The paper reads as if mitochondria are not necessary for tumorigenesis. Please change language to appropriately reflect the role of mitochondria in tumorigenesis and metastasis.

How do they reconcile their data with Raghu Khalluri's data in Nature Cell Biology?

Reviewer #2_System Metabolomics
(Remarks to the Author):

Summary

In addition to biochemical studies focusing on representative in vitro and in vivo models of cancer, we require to understand what is the heterogeneity of metabolism across cancers and how this heterogeneity impacts choice of therapy and patient survival. The work by Gaude & Frezza is a significant contribution in this direction. In particular, the report that mitochondrial metabolism is down regulated in cancers and that it correlates with poor prognosis is in my opinion sufficient to recommend publication in a high impact journal. However, there are some points where the authors should provide additional evidence

Major comments

- 1- Throughout the manuscript there are frequent statements of being up regulated, down regulated, correlated, etc. However, the authors should demonstrate that those associations cannot be obtained just by chance. They should report the statistical significance of those associations.
- 2- Hu et al [1] have previously reported the up regulation of purine and pyrimidine metabolism in cancers. The author's conclusions in this specific aspect are not novel. They should cite that previous work. Interestingly, Hu et al reported no association between TCA cycle and OxPhos and cancers. In contrast, the authors report a down regulation of TCA and OxPhos in cancer, which is one of the key conclusions of this work. The authors should explain why the discrepancy between the report by Hu et al and their results. Is the pruning of promiscuous genes a key methodology to uncover the association between reduces TCA cycle/OxPhos in cancers?
- 3- The observation of up regulation of purine and pyrimidine metabolism in cancers could follow

from the observation of increased proliferation in cancer and the requirement of purines and pyrimidines for cell growth. The authors should determine up to what extent the expression of purine and pyrimidine metabolism gene signatures follow signatures of cell proliferation.

4- Related to point 3, is also the decrease in mitochondrial gene signatures associated with an increase cell proliferation?

References

1. Hu J, Locasale JW, Bielas JH, O'Sullivan J, Sheahan K, Cantley LC, Vander Heiden MG, Vitkup D: Heterogeneity of tumor-induced gene expression changes in the human metabolic network. *Nat Biotechnol* 2013, 31(6):522-U511.

Reviewer #3_Gene regulatory network
(Remarks to the Author):

This manuscript describes an analysis of gene expression data from The Cancer Genome Atlas Project, comprising more than 8000 samples across 20 different cancer entities. To study gene expression in metabolic pathways, the authors compile expression values for genes in these pathways into a pathway score based on gene set enrichment analysis methods. They find a number of metabolic pathways to be dysregulated, including purine biosynthesis, glycolysis, citric acid cycle and oxidative phosphorylation.

Major remarks

1. The authors do not cite and discuss some recent related work, including Gross, Kreisberg & Ideker 2015 (*PLOS ONE* 10:e0142618, 2015) and Reznik & Sander 2015 (*PLOS Comput Biol* 11:e1004176, 2015). They also do not compare their work in real detail with Hu et al. (*Nat Biotechnol* 2013, ref. #4). This needs to be addressed, and the authors should comment on the novelty and originality of their findings compared to already published work.

2. The cellular composition of normal tissues is quite different from tumors, and this may heavily bias gene expression. Also, protein concentrations are known to correlate only poorly with RNA abundance estimates (e.g., Zhang et al. *Nature* 513, 382-387, 2014). This needs to be discussed.

3. Generating P values for each pathway constitutes a multiple testing problem, which needs to be appropriately corrected for. Promiscuity, or association of a particular gene to multiple pathways, presents another statistical challenge; the authors tried to address this by down-weighting their contribution by an ad-hoc approach which, however, is by no means statistically motivated. I highly recommend to consult a professional statistician to get this (and other issues, see below) right.

4. Mixing parametric and non-parametric approaches to estimate correlation (i.e. Pearson's vs Spearman's correlation) will result in non-comparable measures; one should use just one of them.

5. Calculating a mean expression value per pathway is a very crude method to estimate its activity. Consider, e.g., a single gene which is crucial for regulation and changes expression by 2-fold compared to normal tissues. This may happen at low or high levels of the enzyme. A low level will probably not significantly influence the mean value for the pathway while a high level will. Furthermore, this will be more pronounced in small pathways (few other contributions to the mean) than in large pathways, where the effect will be diluted by many other genes. Note that most metabolic pathways are regulated at only few points. It may still be useful to use mean expression per pathway as a first approach, but its limitations need to be discussed.

6. The assignment into "high-survival group" and "low-survival group" is severely flawed. First, by construction the time-to-event or time-to-last-observation intervals overlap. Discriminating by event status will not help since in some cancer entities many samples have short observation

times (early censoring), and patients may have died quickly after the last observation. These patients would, however, be assigned to the "high-survival group" since they have not been known to die of cancer. Second, baseline survival will severely differ between different cancer types, which is, e.g., dramatically short in glioblastoma or pancreatic cancer but multiple times longer in breast cancer or prostate cancer. Finally, TCGA data is severely biased by including multiple therapy regimens (in GBM, e.g. more than 200!), and by notorious incompleteness of clinical data. Thus, survival analysis has to be limited to those cancer entities where a decent fraction has follow-up information, most patients reach sufficient observation times, and therapies are rather similar between them. In addition, the analysis should be done per entity and different cancer types not be mixed.

7. The association with EMT is not directly with the phenotype, which is based on morphology, but with a gene expression signature derived from a cell line experiment. Some researchers even doubt that EMT exists in vivo, and would call it an in-vitro phenotype. The limitations by using an indirect method here should be discussed.

8. Correlation is not causation. Thus, the association of metabolic phenotype with aggressiveness does not indicate that down-regulation of oxidative phosphorylation is actually causing aggressive behavior. Rather, both may be characteristics of an underlying biological state that is characterized by increased metastatic traits but also by reduced energy demands compared to that of primary tumors.

Minor remarks

1. The mathematical notation needs to be corrected; subscripts and superscripts appear in line with the symbols.

2. It is not clear which version of the data has been used. The authors should designate more precisely whether they used "level 3 data", on which transcript model, and add version information to the Supplementary Material. Also, version numbers for the software packages need to be provided.

3. Supplemental Figures 5, 6, and 8 appear to be missing from the PDF document. This may be a system incompatibility using a particular version of Adobe Reader on a Mac, but I'd highly prefer to see the contents that I'm supposed to review.

Response to Reviewers

Reviewers (R): *italic*

Authors (Au): plain text

Reviewer #1_Cancer Metabolism

R: This paper demonstrates that cancers undergo a tissue-specific metabolic rewiring, which converges on downregulation of mitochondrial genes. This is associated with the worst clinical outcome across all cancer types and correlates with induction of epithelial-to-mesenchymal transition (EMT). I think the paper is done well and very important for the field.

Au: We thank the reviewer for this very supportive feedback on our work.

R: I have one issue with the paper in interpretation. I don't think it is appropriate to equate down regulation of mitochondrial genes with mitochondrial dysfunction. It is well known that mitochondrial genes/enzymes are in excess. Thus even 50-75% decrease will not impact oxidative phosphorylation. So please remove that language regarding mitochondrial dysfunction.

Au: We are aware that mitochondrial dysfunction is multifactorial and we agree with the reviewer that appropriate language should be used to avoid misinterpretation of our findings. We have now amended the parts of text where down-regulation of mitochondrial genes was inadvertently linked to mitochondrial dysfunction.

R: Also it is important they clearly state that mitochondria can undergo oxidative and reductive metabolism. Another words, mitochondria always have to engaged in cancer for tumorigenesis. After all mitochondria provides citrate, aspartate and succinyl-CoA necessary for lipid, nucleotide and heme synthesis, respectively. See recent paper in Cell by Ralph Deberardinis. Given this what is the advantage to down regulate these genes? Maybe to

increase or decrease ROS for metastasis? The paper reads as if mitochondria are not necessary for tumorigenesis. Please change language to appropriately reflect the role of mitochondria in tumorigenesis and metastasis.

Au: We fully agree with the reviewer that mitochondrial function is far from dispensable in cancer initiation and progression. We have now expanded the discussion to include a description of the important role of mitochondria in cancer. Please see line 249-268 of the manuscript for details.

R: How do they reconcile their data with Raghu Khalluri's data in Nature Cell Biology?

Au: LeBleu et al (Nat Cell Biol 2014) made two important findings. First, by analyzing the metabolic phenotype of circulating cells and metastasis generated from orthotopic breast cancer, they observed that the expression of OXPHOS genes is increased in circulating tumor cells and it is decreased in metastatic cells, compared to parental cells. Second, they showed that high expression of PGC1 α is associated with decreased survival in a cohort of 161 patients with invasive ductal carcinoma. These results are in partial agreement with our analyses. Indeed, in line with LeBleu et al we also observed a decreased expression of OXPHOS genes in metastatic cells. However, we could not find a significant change in the expression levels of PGC1 α neither between metastatic and primary melanoma cancer patients (BH-p value = 0.37), nor between metastatic and parental 786-O cell lines (BH-p value = 0.51). It is therefore possible that Kalluri's observations are valid in a particular cohort of invasive ductal breast cancers, and the role of PGC1 α in supporting formation of metastasis might be cancer-specific. In line with a possible tissue-specific role of PGC1 α in cancer metastasis, a recent study found that down-regulation of PGC1 α is linked with prostate cancer progression and metastasis, and genetic reactivation of PGC1 α suppresses the formation of prostate cancer metastases (Torrano et al., 2016). We have now included a discussion of these findings in our manuscript (please see lines 281-294 of the manuscript for details).

Reviewer #2_System Metabolomics

R: Summary. In addition to biochemical studies focusing on representative in vitro and in vivo models of cancer, we require to understand what is the heterogeneity of metabolism

across cancers and how this heterogeneity impacts choice of therapy and patient survival. The work by Gaude & Frezza is a significant contribution in this direction. In particular, the report that mitochondrial metabolism is down regulated in cancers and that it correlates with poor prognosis is in my opinion sufficient to recommend publication in a high impact journal.

Au: We thank the reviewer for appreciating the impact of our work.

R: However, there are some points where the authors should provide additional evidence

Major comments

1- Throughout the manuscript there are frequent statements of being up regulated, down regulated, correlated, etc. However, the authors should demonstrate that those associations cannot be obtained just by chance. They should report the statistical significance of those associations.

Au: We are aware of the importance of accounting for random (or chance) variation in statistics and indeed our analyses were performed applying stringent statistical tests. For instance, differential expression of metabolic genes was determined by applying Wald's statistical test and p-values were corrected for multiple testing with Benjamini-Hochberg method. A false discovery rate (FDR) of 5% was applied by considering Benjamini-Hochberg-corrected p-values < 0.05 as statistically significant. To determine up- or down-regulation of metabolic pathways gene set enrichment analysis (GSEA) was performed. Enrichment p-values were corrected for multiple testing with Benjamini-Hochberg correction method. When correlation was used to determine the relationship between variables, Spearman method was applied and p-values were calculated. As above, p-values were corrected with Benjamini-Hochberg method to account for type 1 error. Details of all statistical tests and analyses are reported in the methods section. P-values of tests applied (Wald's statistics, GSEA), as well as correlation coefficients and p-values, are included as supplementary material.

R: 2- Hu et al [1] have previously reported the up regulation of purine and pyrimidine metabolism in cancers. The author's conclusions in this specific aspect are not novel. They should cite that previous work. Interestingly, Hu et al reported no association between TCA cycle and OxPhos and cancers. In contrast, the authors report a down regulation of TCA and

OxPhos in cancer, which is one of the key conclusions of this work. The authors should explain why the discrepancy between the report by Hu et al and their results. Is the pruning of promiscuous genes a key methodology to uncover the association between reduces TCA cycle/OxPhos in cancers?

Au: We have now expanded the discussion on the work of Hu *et al*, which was already cited in our manuscript, and explained more in details the differences and similarities between this work and our results (please see lines 224-233 for details). The reviewer hypothesizes that the different behavior of TCA cycle and OXPHOS between our study and Hu *et al*'s is caused by pruning promiscuous genes. However, it is worth noting that OXPHOS ranked in our top-altered metabolic pathways both in the non-corrected and promiscuity-corrected analyses (see Supplementary Figure 2). We think that the discrepancy between Hu *et al* and our work is due to differences in the metabolic gene set definition: while Hu *et al* applied the structure of metabolic pathways offered by the Kyoto Encyclopedia of Genes and Genomes (KEGG), we manually curated metabolic gene sets. More in detail, we used metabolic genes and pathways as defined in the genome-scale human reconstruction of metabolic network Recon1 (Duarte et al., 2007) and we manually curated the assignment of genes into pathways by referring not only to KEGG, but also to the Human Metabolome DataBase (HMDB) and OMIM libraries, as well as by referring to several studies in the literature. In some cases, this led to the definition of different metabolic gene sets. For instance, while the TCA cycle is composed of 30 genes according to KEGG database, our manually curated gene set included 58 genes as part of the TCA cycle. Our assignment of metabolic genes into pathways is provided as supplementary material.

R: 3- The observation of up regulation of purine and pyrimidine metabolism in cancers could follow from the observation of increased proliferation in cancer and the requirement of purines and pyrimidines for cell growth. The authors should determine up to what extent the expression of purine and pyrimidine metabolism gene signatures follow signatures of cell proliferation.

4- Related to point 3, is also the decrease in mitochondrial gene signatures associated with an increase cell proliferation?

Au: In the original version of our manuscript we investigated the link between nucleotide synthesis and proliferation of cancer cells by assessing the correlation between expression of

purine biosynthesis pathway and growth rate of the NCI-60 panel of cancer cell lines (Supplementary Figure 3 and lines 93-96, main text). We found a highly significant and positive correlation between expression of purine biosynthesis and proliferation of cancer cells (Supplementary Figure 3a). Of note, we also found a very strong association between cancer cell growth and *PAICS*, the gene of purine biosynthesis most commonly up-regulated among cancers (Supplementary Figure 3b). Finally, we also observed a positive ($r^2=0.36$) and near-significance (p-value=0.051) correlation of pyrimidine biosynthesis with cell growth of the NCI-60 cell lines.

We also assessed the correlation of mitochondrial pathways with cell proliferation of the NCI-60 panel of cancer cell lines. Surprisingly, we observed that both TCA cycle and OXPHOS are positively correlated ($r^2=0.42$ and 0.39 , respectively) and significant (p-value=0.02 and 0.04, respectively) with cancer cells proliferation. Such discrepancy might be due to the fact that the NCI-60 cell lines are grown *in vitro*, where metabolic requirements are different from those observed *in vivo*, where the great majority of the analyses were carried out.

Reviewer #3_Gene regulatory network

R: This manuscript describes an analysis of gene expression data from The Cancer Genome Atlas Project, comprising more than 8000 samples across 20 different cancer entities. To study gene expression in metabolic pathways, the authors compile expression values for genes in these pathways into a pathway score based on gene set enrichment analysis methods. They find a number of metabolic pathways to be dysregulated, including purine biosynthesis, glycolysis, citric acid cycle and oxidative phosphorylation.

Major remarks

1. The authors do not cite and discuss some recent related work, including Gross, Kreisberg & Ideker 2015 (PLOS ONE 10:e0142618, 2015) and Reznik & Sander 2015 (PLOS Comput Biol 11:e1004176, 2015). They also do not compare their work in real detail with Hu et al. (Nat Biotechnol 2013, ref. #4). This needs to be addressed, and the authors should comment on the novelty and originality of their findings compared to already published work.

Au: We thank the reviewer for pointing out these papers. We have expanded the discussion to include these relevant studies and compared our findings to their results (please see line 223-237 of our manuscript for details).

R: 2. The cellular composition of normal tissues is quite different from tumors, and this may heavily bias gene expression. Also, protein concentrations are known to correlate only poorly with RNA abundance estimates (e.g., Zhang et al. Nature 513, 382-387, 2014). This needs to be discussed.

Au: We agree with the reviewer that the comparison of tumor and normal samples presents several caveats, of which we are aware. Qualitative and quantitative differences in cellular composition can exist between cancer and normal tissues, among which infiltration of immune cells, as well as other blood-derived cells, are well known. Indeed, we were impressed to observe that, in line with Hu *et al*, expression of metabolic genes in cancer samples was highly reminiscent of their tissue of origin, when compared to other tissues. These results indicate that, despite differences in cellularity, cancer samples maintain the metabolic identity of their tissue of origin and suggest that differences in cell infiltrates might contribute to background noise variation between samples.

We agree with the reviewer that estimating protein concentrations from abundance of RNA transcript levels is not devoid of assumptions. We have discussed these limitations in line 295-299 of the manuscript.

R: 3. Generating P values for each pathway constitutes a multiple testing problem, which needs to be appropriately corrected for. Promiscuity, or association of a particular gene to multiple pathways, presents another statistical challenge; the authors tried to address this by down-weighting their contribution by an ad-hoc approach which, however, is by no means statistically motivated. I highly recommend to consult a professional statistician to get this (and other issues, see below) right.

Au: The reviewer raises concerns about the strength of our analyses. Our analyses underwent extensive statistical validation. For instance, to determine up- or down-regulation of metabolic pathways we performed gene set enrichment analysis (GSEA). Enrichment p-values were corrected for multiple testing with Benjamini-Hochberg correction method and a false discovery rate (FDR) of 5% was applied on Benjamini-Hochberg-corrected p-values.

For the correction of promiscuous genes, we applied a heuristic method designed for the *ad-hoc* down-weighting of promiscuous genes to account for metabolic pathways that are composed predominantly by promiscuous genes. A similar approach for down-weighting

promiscuous genes was proposed by Tarca and colleagues (Tarca et al., 2012), where they used the frequency of each gene across gene sets to construct a score for down-weighting genes statistics in enrichment analysis. Our approach presents a more stringent down-weighting applied on t-statistics. Of note, this approach was discussed with a professional statistician, who is now formally acknowledged in our manuscript.

R: 4. Mixing parametric and non-parametric approaches to estimate correlation (i.e. Pearson's vs Spearman's correlation) will result in non-comparable measures; one should use just one of them.

Au: We thank the reviewer for raising this concern. We have re-calculated the relevant correlation analyses by uniformly applying Spearman's method. New results are now displayed in the manuscript:

- Fig. 2a: correlation analysis of metabolic gene expression between normal and cancer samples.
- Fig. 3c: correlation of mean OXPHOS values with mean EMT values.
- Supplementary Table 6: correlation coefficients and p-values of metabolic gene expression between normal and cancer samples.
- Supplementary Figure 4: correlation of mean pathway expression with metabolite abundance in breast cancer patients.
- Supplementary Figure 8: scatter plots of mean OXPHOS vs mean EMT values for each cancer type.

We amended the results section in order to display new results obtained (lines 126 and 129), as well as we updated methods and figure legends accordingly. Importantly, this new analysis did not affect our final conclusions.

R: 5. Calculating a mean expression value per pathway is a very crude method to estimate its activity. Consider, e.g., a single gene which is crucial for regulation and changes expression by 2-fold compared to normal tissues. This may happen at low or high levels of the enzyme. A low level will probably not significantly influence the mean value for the pathway while a high level will. Furthermore, this will be more pronounced in small pathways (few other contributions to the mean) than in large pathways, where the effect will be diluted by many other genes. Note that most metabolic pathways are regulated at only few points. It may still be useful to use mean expression per pathway as a first approach, but its limitations need to

be discussed.

Au: We are aware of the limitations of using mean pathway expression to investigate pathway activity and the reviewer raises a valid concern. We accounted for differential effects of low and high expressed genes by applying *variance stabilizing transformation* (Huber et al., 2002), an approach to manage the dependence of variance of an intensity from its mean. As explained in our methods, mean expression of pathways was calculated on variance-stabilized gene intensities.

The reviewer raised a valid concern about the effect of pathway size on determining intra-pathway variability of gene expression changes. To address this concern, we assessed whether pathway size influences the dispersion of gene expression changes. We could not find a strong link between pathway size and interquartile range of gene fold change between tumor and normal samples (please see **Figure 1 for reviewer only** for a representative example), suggesting that the size of the pathway is not a strong determinant of mean pathway expression. Yet, more work is required to dissect to what extent each individual enzyme contributes to the overall function of the pathway. The limitations of our approach have been described in the discussion section of our manuscript (lines 300-302 for details)

Figure 1 for Reviewer only. Effect of pathway size on intra-pathway dispersion of gene expression changes. Number of genes in each pathway (x axis) is plotted against the interquartile range of mean gene fold changes (y axis) between tumor and normal samples, for BRCA, GBM, KIRP and PRAD datasets.

R: 6. The assignment into "high-survival group" and "low-survival group" is severely flawed. First, by construction the time-to-event or time-to-last-observation intervals overlap. Discriminating by event status will not help since in some cancer entities many samples have short observation times (early censoring), and patients may have died quickly after the last observation. These patients would, however, be assigned to the "high-survival group" since they have not been known to die of cancer. Second, baseline survival will severely differ between different cancer types, which is, e.g., dramatically short in glioblastoma or pancreatic cancer but multiple times longer in breast cancer or prostate cancer. Finally, TCGA data is severely biased by including multiple therapy regimens (in GBM, e.g. more than 200!), and by notorious incompleteness of clinical data. Thus, survival analysis has to be limited to those cancer entities where a decent fraction has follow-up information, most patients reach sufficient observation times, and therapies are rather similar between them. In addition, the analysis should be done per entity and different cancer types not be mixed.

Au: The criticisms raised by the reviewer on our survival analyses are not fully justified and indicate that some clarification to our method of analysis of patient's survival is needed. Although the definition of "Low" and "High survival" groups is based on event status, we designed the analyses to take into account early censoring. The "High survival" group includes only patients that have been censored alive for an amount of time greater than or equal to the 75th percentile of the "Low survival" observation time. This method allowed us to filter out patients with short observation times based on the cancer type-specific window of observation. We expanded the methods section (lines 386-396) by explaining more in detail the definition of "High" and "Low survival" patient groups and by showing how this method applies to the BRCA dataset as an example. Importantly, "High" and "Low survival" groups were defined for each cancer type individually, which allowed us to account for differences in baseline survival time. Metabolic pathways significantly enriched from the analysis of each cancer type individually were then pooled together to find metabolic pathways that are changed between "High" and "Low survival" patients in at least 25% of cancer types.

As noticed by the reviewer the integration of treatment regimen information into survival analysis of TCGA data can be a daunting task. Despite the important effects that therapeutic treatments can have on survival of cancer patients, subdividing patients would result in very small number being tested (or in patient-specific cases), thus impinging on statistical power of the analysis. Based on these considerations, we decided that accounting for therapeutic treatment in our analysis of "High" vs "Low survival" patients would lead to over-fitting and,

as such, we assumed it would be confined to background noise. The integration of therapeutic treatments into survival analysis would require a thorough investigation of several statistical techniques, beyond the scope of this manuscript.

R: 7. The association with EMT is not directly with the phenotype, which is based on morphology, but with a gene expression signature derived from a cell line experiment. Some researchers even doubt that EMT exists in vivo, and would call it an in-vitro phenotype. The limitations by using an indirect method here should be discussed.

Au: We feel this reviewer's comment might derive from a misunderstanding of our analyses. The association of poor patient survival with EMT was obtained by considering gene expression profiles of human cancer samples, it was not obtained from *in vitro* experiments. The reviewer raises also some concerns about the relevance of EMT in cancer metastasis *in vivo*. To corroborate this aspect of our work, we performed GSEA on metastatic vs primary melanoma cancer samples and found that EMT is strongly up-regulated in metastatic samples. We validated these results on metastatic vs parental 786-O cell lines and found that EMT is highly induced in the metastatic counterpart. Furthermore, we have replaced the enrichment analyses of OXPHOS genes in metastatic 786-O cells with a volcano plot to further support the claim that this pathway is the most downregulated metabolic pathway in these cells (New Fig. 4b). Although these results do not prove that EMT drives metastasis, they further corroborate the association between suppression of OXPHOS genes, presence of EMT signature and metastasis. We have now included these observations in Supplementary Figure 9 and in line 184/185 and 196/197 of the results section.

R: 8. Correlation is not causation. Thus, the association of metabolic phenotype with aggressiveness does not indicate that down-regulation of oxidative phosphorylation is actually causing aggressive behavior. Rather, both may be characteristics of an underlying biological state that is characterized by increased metastatic traits but also by reduced energy demands compared to that of primary tumors.

Au: We agree with the reviewer on the limitations and potential pitfalls of drawing conclusions from correlation analyses. We believe that co-variation of different factors can help in generating hypotheses that, despite their statistical robustness, have to be validated with appropriate experimental work in order to establish a more direct causal link. Given the

importance and potential impact of these assumptions we expanded the discussion in order to account for the limitations of our approach (line 302-307) Moreover, we corrected the language in the results section where correlation results were inadvertently linked to mechanistic conclusions.

R: Minor remarks

1. The mathematical notation needs to be corrected; subscripts and superscripts appear in line with the symbols.

Au: We have now corrected the mathematical notation.

R: 2. It is not clear which version of the data has been used. The authors should designate more precisely whether they used "level 3 data", on which transcript model, and add version information to the Supplementary Material. Also, version numbers for the software packages need to be provided.

Au: Detailed information about the level and type of RNAseq data used, as well as version numbers of software and packages, are now included in the methods section.

R: 3. Supplemental Figures 5, 6, and 8 appear to be missing from the PDF document. This may be a system incompatibility using a particular version of Adobe Reader on a Mac, but I'd highly prefer to see the contents that I'm supposed to review.

Au: We were unaware of any problem with the file formats sent to reviewers and understand the frustration of not having the chance to fully access the material object of revision. We are disappointed as much as this reviewer to discover that the process of file reformatting failed to provide a complete version of the material we submitted. Other reviewers did not mention this problem, so it might be a platform-specific issue.

Reviewers' comments:

Reviewer #1 (Remarks to the Author):

The paper is acceptable!

Reviewer #2 (Remarks to the Author):

The authors have provided a satisfactory response to my previous comments. This work is an important contribution to our understanding of cancer metabolism that supports its publication in Nature Communications.

Reviewer #3 (Remarks to the Author):

I appreciate that the authors have significantly improved the manuscript and addressed most of my points.

The most important original finding of the authors compared to previous studies is the association of metabolic pathways, in particular primary energy metabolism in the mitochondria, with clinical aggressiveness, survival and metastatic phenotype. I still have some concerns with respect to the analysis that the authors carried out.

1. I could not directly confirm the correctness of the underlying gene expression values. Mean expression as well as log₂-fold change and log₂-standard deviation are given in Supplementary Table 2. I was struck by the unexpectedly high mean counts for most genes (many above 1000), which follow an unusual distribution. The authors state in the manuscript that they used RSEM abundance estimates as provided by TCGA. I used the cbio portal (<http://cbioportal.org>), which hosts the very same values from the very same sets. In the attachment, I provide three exemplary plots generated from this portal. These are co-expression plots, but the only relevant dimension here is on the x-axis, which has RNA RSEM expression values for the very same set, PCPG (pheochromocytoma and paraganglioma). As can be seen for A4GALT, the values on the cbio portal range from 0 to about 1800, with an approximate mean expression of 200. The authors report 3116.05 as "baseMean" in their supplementary table. For A4GNT, the values range from 0 to 7, with an approximate mean somewhere below 1. The authors, on the contrary, report 1219.8 as "baseMean". For ABCB11, the values on the cbio portal range from 0 to 6 (mean below 1) while the authors report 116.4. Neither the absolute values nor their relation are anywhere close to the values from cbio.

2. The authors seem to not always follow their own method to define "high survival" and "low survival" groups. Briefly, they take the 75th percentile of the survival times for the patients with status "dead" and declare patients with status "alive" as members of the "high survival" group if their survival time is greater than this percentile. That means that, for patients in the "low survival" group, the time where the KM estimate curve crosses 0.25 should be an absolute boundary for observation times in the "high survival" group. Indeed, that is what can be observed for most cancer types in Supplementary Figure 6. However, for some cancer types (i.e. HNSC, KIRP, LUAD, LUSC, STAD, and UCEC) there are many patients in the "high survival" group which have been censored at substantially shorter observation times. I provide an example for LUAD, where the horizontal green bar at 0.25 on the y axis marks the fraction of 25% survival in the "low survival" group (somewhere between 3 and 4 years). However, almost all patients in the "high survival" group have been censored at shorted times (within the light blue area). In fact, there are only four patients for which observation time is sufficiently high - these are marked by crosses (censoring) on the black survival curve outside of the blue area. Given the poor prognosis for most lung adenocarcinoma patients, a substantial fraction of patients with short observation times who

are currently designated "alive" may have died later and thus should be in fact in the "low survival" group. Consequently, any difference in gene expression for metabolic genes in these groups is likely not attributable to prognosis or survival. I would thus recommend that analysis for the cancer types mentioned is re-calculated using appropriate definition of groups, with the constraint that analysis should not be done if groups get too small (e.g., 4 high survivors versus about 100 low survivors does not make really sense).

3. Even if the method is followed correctly, it will create biases. Let me take pancreatic adenocarcinoma as an example. This cancer type has an extremely bad prognosis, with 5-year survival rates way below 5%. I have re-calculated the survival curves according to the authors' definition (see attached figure). The samples on the red part of the survival curves are excluded ("alive", but survival time below 75th percentile, which is 596.5 days). The samples on the green part of the curve are in the "high survival" group. But, from the blue curve it is obvious that many (if not virtually all) of the remaining patients will die between 600 and 1500 days, thus you'd expect that would also apply to the patients designated "alive" in the high survival group with observation time in this range. This again will significantly distort the analysis of differential gene expression as only 7 or so of the originally 38 patients in the "high survivor" group have really survived and thus had a significantly better prognosis than those in the "low survivor" group. This reasoning will not apply to all cancer types equally as it depends on the prior probability of surviving at a given time (here: at the threshold for the "high survival" group), or in other words on which level the KM estimate curve approximates at very long times (lower bound). To avoid these biases, I propose to add a different type of analysis. Gene expression values for the respective pathways should be used to discriminate patients for each cancer type into different groups (e.g., quartiles). Then, survival fits can be performed for each of these groups, and the extreme quartiles can be compared by a log-rank test. This way, the biases that may be in the recording of follow-up can be somehow avoided, under the assumption that they distribute similarly in the different groups defined by metabolic pathway gene expression.

Gaude and Frezza
NCOMMS-16-05237

Response to Reviewers

Reviewers (R): *italic*

Authors (Au): plain text

Reviewer #1:

R: The paper is acceptable!

Au: Thank you for appreciating the revised version of the manuscript

Reviewer #2:

R: The authors have provided a satisfactory response to my previous comments. This work is an important contribution to our understanding of cancer metabolism that supports its publication in Nature Communications.

Au: Thanks for appreciating the revised version of the manuscript and for emphasizing the relevance of our work.

Reviewer #3:

R: I appreciate that the authors have significantly improved the manuscript and addressed most of my points. The most important original finding of the authors compared to previous studies is the association of metabolic pathways, in particular primary energy metabolism in the mitochondria, with clinical aggressiveness, survival and metastatic phenotype.

Au: We thank the reviewer for appreciating the improvements of the revised version of the manuscript and for highlighting the novelty of our work.

R: I still have some concerns with respect to the analysis that the authors carried out. I could not directly confirm the correctness of the underlying gene expression values. Mean expression as well as log2-fold change and log2-standard deviation are given in Supplementary Table 2. I was struck by the unexpectedly high mean counts for most genes (many above 1000), which follow an unusual distribution. The authors state in the manuscript that they used RSEM abundance estimates as provided by TCGA. I used the cbio portal (<http://cbioportal.org>), which hosts the very same values from the very same sets. In the attachment, I provide three exemplary plots generated from this portal. These are co-expression plots, but the only relevant dimension here is on the x-axis, which has RNA RSEM expression values for the very same set, PCPG (pheochromocytoma and paraganglioma). As can be seen for A4GALT, the values on the cbio portal range from 0 to about 1800, with an approximate mean expression of 200. The authors report 3116.05 as "baseMean" in their supplementary table. For A4GNT, the values range from 0 to 7, with an approximate mean somewhere below 1. The authors, on the contrary, report 1219.8 as "baseMean". For ABCB11, the values on the cbio portal range from 0 to 6 (mean below 1) while the authors report 116.4. Neither the absolute values nor their relation are anywhere close to the values from cbio.

A: We thank the reviewer for thoroughly checking our analytical pipeline. We wish to clarify that the values used for gene expression analysis are mRNA Read Counts and not RSEM values, as wrongly indicated in the method section. We apologize for the confusion generated by this error, which has been now corrected (line 328). Furthermore, we would like to point out that the presented data have been processed using the R package DEseq2 (as described in methods). This step is crucial for the reproduction of our analyses.

R: The authors seem to not always follow their own method to define "high survival" and "low survival" groups. Briefly, they take the 75th percentile of the survival times for the patients with status "dead" and declare patients with status "alive" as members of the "high survival" group if their survival time is greater than this percentile. That means that, for patients in the "low survival" group, the time where the KM estimate curve crosses 0.25 should be an absolute boundary for observation

times in the "high survival" group. Indeed, that is what can be observed for most cancer types in Supplementary Figure 6. However, for some cancer types (i.e. HNSC, KIRP, LUAD, LUSC, STAD, and UCEC) there are many patients in the "high survival" group which have been censored at substantially shorter observation times. I provide an example for LUAD, where the horizontal green bar at 0.25 on the y axis marks the fraction of 25% survival in the "low survival" group (somewhere between 3 and 4 years). However, almost all patients in the "high survival" group have been censored at shorted times (within the light blue area). In fact, there are only four patients for which observation time is sufficiently high - these are marked by crosses (censoring) on the black survival curve outside of the blue area. Given the poor prognosis for most lung adenocarcinoma patients, a substantial fraction of patients with short observation times who are currently designated "alive" may have died later and thus should be in fact in the "low survival" group. Consequently, any difference in gene expression for metabolic genes in these groups is likely not attributable to prognosis or survival. I would thus recommend that analysis for the cancer types mentioned is re-calculated using appropriate definition of groups, with the constraint that analysis should not be done if groups get too small (e.g., 4 high survivors versus about 100 low survivors does not make really sense).

Even if the method is followed correctly, it will create biases. Let me take pancreatic adenocarcinoma as an example. This cancer type has an extremely bad prognosis, with 5-year survival rates way below 5%. I have re-calculated the survival curves according to the authors' definition (see attached figure). The samples on the red part of the survival curves are excluded ("alive", but survival time below 75th percentile, which is 596.5 days). The samples on the green part of the curve are in the "high survival" group. But, from the blue curve it is obvious that many (if not virtually all) of the remaining patients will die between 600 and 1500 days, thus you'd expect that would also apply to the patients designated "alive" in the high survival group with observation time in this range. This again will significantly distort the analysis of differential gene expression as only 7 or so of the originally 38 patients in the "high survivor" group have really survived and thus had a significantly better prognosis than those in the "low survivor" group. This reasoning will not apply to all cancer types equally as it depends on the prior probability of surviving at a given time (here: at the threshold for the "high survival" group), or in other words on which level the

KM estimate curve approximates at very long times (lower bound). To avoid these biases, I propose to add a different type of analysis. Gene expression values for the respective pathways should be used to discriminate patients for each cancer type into different groups (e.g., quartiles). Then, survival fits can be performed for each of these groups, and the extreme quartiles can be compared by a log-rank test. This way, the biases that may be in the recording of follow-up can be somehow avoided, under the assumption that they distribute similarly in the different groups defined by metabolic pathway gene expression.

A: We thank the reviewer for highlighting potential bias generated by the definition of High and Low Survival groups. Since this is an critical aspect of our work, we have comprehensively revised our analyses taking into account referee's suggestions. We first performed the analysis suggested by the reviewer by subdividing patients into quartiles of mean pathway expression for each cancer type. We then estimated the effect of pathway expression on overall patient survival. With this analysis we found that whilst some metabolic pathways were linked to patient survival, the associations were very weak and not widespread across different cancer types. We hypothesize that lack of strong correlations is due to the fact that mean pathway expression is not a strong predictor of metabolic signature, as also pointed out by this referee in a previous comment. To overcome this issues and strengthen our analyses, we re-designed the definition of High and Low survival groups and applied metabolic GSEA to these groups. For each cancer type we have included in the High Survival group patients that have been censored alive for longer than the 75th percentile of the total duration of the follow-up study. On the other hand, the Low Survival group included patients that have died within the 75th percentile of the total follow-up study duration (see Figure 1 for reviewer only for a graphical representation of this classification). New Supplementary Figure 6 has been updated to show survival curves for each cancer type based on this improved classification.

Of note, this new method avoids overlap between High and Low survival groups and potential biases that could arise from short follow-up recordings, as pointed out by the referee. This new classification affected group size of High and Low survival patients for each cancer type, resulting in some groups being poorly represented. We followed the reviewer’s suggestion and excluded from gene expression analysis those cancer types where the High or Low survival group were smaller than 5 patients. Cancer types excluded were CHOL, PCPG, PRAD, READ, THCA. Importantly, OXPHOS was confirmed as the most down-regulated pathway in Low Survival patients compared to the High survival group, demonstrating the robustness of our findings even under more stringent conditions. Figure 3a and 3b have been amended to display new results, and Results (lines 166-173) and Methods (lines 376-395) sections have been updated with new findings and procedures, respectively. We have also included results of this new gene expression analysis, together with each group’s size, in Supplementary Table 9.

Reviewer #3 (Remarks to the Author):

I appreciate the additional work of the authors, which addresses all of my remaining points.

Gaude and Frezza
NCOMMS-16-05237B

Response to Reviewers

Reviewers (R): *italic*
Authors (Au): plain text

Reviewer #3:

R: I appreciate the additional work of the authors, which addresses all of my remaining points.

Au: Thank you for appreciating the revised version of the manuscript